# Hospital Restraints: Safe or Dangerous? A Case of Hospital Death Due to Asphyxia from the Use of Mechanical Restraints

**DOI:** 10.3390/ijerph19148432

**Published:** 2022-07-10

**Authors:** Carmen Scalise, Fabrizio Cordasco, Matteo Antonio Sacco, Valerio Riccardo Aquila, Pietrantonio Ricci, Isabella Aquila

**Affiliations:** 1Institute of Legal Medicine, Department of Medical and Surgical Sciences, University “Magna Graecia” of Catanzaro, 88100 Catanzaro, Italy; scalisecarmen@libero.it (C.S.); cordasco@unicz.it (F.C.); matteoantoniosacco@gmail.com (M.A.S.); ricci@unicz.it (P.R.); 2Department of Medical and Surgical Sciences, University “Magna Graecia” of Catanzaro, 88100 Catanzaro, Italy; valerio_aquila@hotmail.it

**Keywords:** forensic science, means of restraint, asphyxia, forensic autopsy, surveillance, accidental death

## Abstract

Asphyxia can be defined as an impediment to the influx of air into the respiratory tract, leading to tissue hypoxia. By restraint, we mean the use of physical, pharmacological and/or environmental means to limit the subject’s ability to move. Fall prevention is the main reason restraint is used. Unfortunately, restraint can sometimes be fatal. There are few studies in the literature on this subject. We report the case of a man with Down syndrome in a psychiatric clinic found dead between the bed and the floor of the room where he was hospitalized. The analysis of the scene showed the presence of a means of a restraint, located around the man’s chest and neck, which kept him tied to the bed and applied a constricting mechanical action. There was doubt as to the cause of death. For this reason, an inspection of the scene and an autopsy were carried out. Upon opening the chest, blood infiltration of the left intercostal muscles that was topographically compatible with external cutaneous excoriation (sign of restraint) became evident. In view of the danger of using restraint, it is necessary to evaluate the means of restraint as an extraordinary and not an ordinary procedure in patient management. Each patient undergoing restraint measures must be carefully monitored by specialized personnel. Greater surveillance of the nurse/patient ratio is necessary to reduce the use of restraints. In this case report, we highlight the lack of surveillance of patients subjected to restraint.

## 1. Introduction

Asphyxia can be defined as an impediment to the influx of air into the respiratory tract and, consequently, tissue hypoxia [1]. In addition, it is made up of two further mechanical factors: the compression of the nerve and vascular bundles of the neck. Strangulation, in particular, is caused by the compression of the neck by means of a lace or other suitable means. In strangulation, the first factor involved is neural, determined by the compression and stimulation of the vagus nerve. Asphyxiated death can occur due to suicide, murder, or accidents. Sometimes, the boundary between the three modes of death is not well defined. 

Asphyxiation is often described in the deaths of people with mental illnesses. In these cases, often during psychic decompensation, physical restraint is applied. The restraint is applied above all in subjects in states of extreme agitation or delirium, combined with an aggressive attitude. The application of restraint is found in various areas. In recent months, the deaths of subjects who are physically constrained at the time of death and, in particular, by police personnel have become prominent. Recently, EMF Strommer et al. analyzed the famous case of the American George Floyd, focusing in particular on restraint-related asphyxia as a probable cause of death [2].

Our work aims to focus attention on accidental asphyxiation deaths and, in particular, on means of restraint. The means of restraint, which are used increasingly often in hospitals and nursing homes as a means of protection for patients in order to prevent falls or violent behavior, can themselves become dangerous for patients if they are not properly applied.

The purpose of this work is to demonstrate how death from asphyxiation by means of physical restraint cannot always be defined as accidental. We also propose an operational protocol to reduce the risk of fatal events. Investigators need to analyze how the health surveillance of patients undergoing physical restraint is performed. Inadequate surveillance could herald death by asphyxiation, which cannot always be defined as accidental.

## 2. Materials and Methods

First, a literature review was carried out by entering the keywords “asphyxia” and “means of restraint” into the PubMed NCBI search engine. A detailed study of the medical records was carried out, specifically on circumstances relating to the decision to use restraint devices. An inspection of the room was carried out with photographic surveys and measurements of the restraint device found with respect to the size of the bed, the distance from the ground and the body. External examination of the corpse was carried out with photographic surveys and measurements of the excoriations found. An autopsy was then performed with a macroscopic examination of the organs. Each organ was examined, photographed and preserved in formaldehyde for subsequent histological examination with hematoxylin and eosin staining and slide preparations for subsequent microscopic analysis. Toxicological investigations were carried out on biological fluids collected during the autopsy (blood, vitreous humor, urine, bile) with the immunoenzymatic method on the ILAB 600 device.

## 3. Case Report

We report the case of a 60-year-old man with Down syndrome in a psychiatric clinic found dead between the bed and the floor of the room where he was hospitalized. A special means of restraint was applied to the victim to suppress episodes of nocturnal agitation. The means used consisted of a non-elastic belt (Figure 1). In addition, the belt had several locks and could only be opened with a magnet key. Considering the location of the victim, who was found with a means of restraint that kept him tied to the bed, at the time of the discovery of his body, the investigators had many doubts as to the circumstances and manner of death, which they suspected had been violent. The analysis of the testimonies of the nurses on duty raised doubts because it was not clear how the belt worked and how it was used by the personnel. 

## 4. Results

### 4.1. Medical-Record Investigation

A study was carried out on the medical records seized during an inspection by the investigators. Episodes of lipothymia with hypotension and bradycardia were reported in the victim’s medical record. The medical record reported the drug therapy administered to the victim and the need for the use of a restraint was reported. In particular, the restraint was performed with a safety belt for the bed, used in critical hours, especially at night, and authorized by a family member, due to the victim’s motor agitation.

### 4.2. Crime-Scene Investigation

The investigators carried out a judicial inspection of the clinic. The analysis of the scene showed the presence of a means of restraint that kept the man tied to the bed. This means was located around the man’s chest and neck, applying a mechanical constricting action on the anatomical areas described. The scene was examined by investigating the use of other restraint devices in other rooms of the clinic and on other patients. The investigators seized and analyzed the means of restraint used, determining the size of the vehicle and reproducing its closure. The data were analyzed and compared with others.

### 4.3. Autopsy and Histopathological Findings

At the autopsy, an analysis of the lesions present on the corpse was carried out and a measurement of the diameter of the chest and abdomen was performed to verify their compatibility with the means of restraint used, which had a magnetic closure with a rigid, non-elastic band and a double closure with extendable plastic hooks (Figure 2). In particular, the analysis of the vehicle showed the presence of a restraint belt 115 cm long and 12.5 cm wide (Figure 1 and Figure 2). 

This belt was not elastic and had buttonholes; outside these slots, there were two other means of containment with plastic closure, as well as a special metal device with magnetic closure (Figure 2 and Figure 3).

In particular, the compatibility of the metal device with a magnetic closure associated with the restraint device was evaluated and ascertained. It was also ascertained that the metal device could only be opened with the application of a magnet that featured a specific green key ring (Figure 3). It was therefore assumed that this restraint belt with these means was further closed by means of the metal device and by means of the aforementioned magnetic closure mechanism. The external examination of the victim showed the presence of hypostatic spots on the hands and feet (glove and sock) and abrasions on the left chest, left elbow and back. In addition, the presence of sub-conjunctival petechiae was found. At the opening of the thorax, there was evidence of blood infiltration of the left intercostal muscles, which was topographically compatible with external cutaneous excoriation (sign of restraint) (Figure 4, Figure 5 and Figure 6). The autopsy also showed signs of compression of the thorax and of the left side of the neck, with left-vagus-nerve injury, although no traumatic injury was found on the phrenic nerve. 

Both lungs showed the presence of sub-pleural petechiae. Petechiae were also present at the level of the right cardiac auricle. The lungs, crackling to the touch as from pulmonary emphysema, showed rupture of the pulmonary septa as from mechanical respiratory efforts in response to asphyxiation. Nothing pathological was found in the other thoracic organs, head or abdomen. Furthermore, no particular data emerged from the histo-pathological (except for signs of pulmonary emphysema with edema) and toxicological examination.

## 5. Discussion

By restraint, we mean the use of physical, pharmacological and/or environmental means to limit the ability of a subject to move [3]. Physical restraint has a prevalence of 15.8% in hospitals and 68.7% in RCFs (residential care facilities) [4]. Fall prevention is the main reason restraint is used. The main physical restraints used in healthcare facilities are bed rails, belts, geriatric tables and other mechanical devices [5]. However, some studies have shown the ineffectiveness of these means in preventing falls; indeed, they cause death in patients due to asphyxiation or trauma [6,7,8]. In fact, there are several harmful consequences of this form of restriction. Patients can develop pressure ulcers, edema, contractures, breathing problems and death from strangulation or fractures [9]. Physical restraints are still used, especially in disoriented and dementia patients, as well as in agitated and violent patients, to prevent falls, to make it easier for health professionals to control patients, to prevent the interruption of a treatment and to overcome problems associated with staff shortages [10,11]. The literature has also shown the risk related to the seclusion and restraint of subjects affected by mental illness. Various studies have focused on the risk of suicide and accidents, especially in prisoners or people in correctional institutions affected by psychiatric illnesses with multiple risk factors. Educational programs with suicide-risk evaluation are recommended in these contexts [12,13,14,15]. In Italy, articles n. 13 and n. 32 of the Constitution clearly state that no form of restriction of personal freedom or compulsory treatment is allowed except by law and out of respect of human dignity [16]. Therefore, restraints must be considered only in exceptional circumstances, such as emergency situations, in which patients demonstrate that represents an immediate danger to them or to others. 

In the case reported, we point out that the use of a restraint caused the victim to die from strangulation. In particular, it emerged from the forensic investigations that all the patients in the clinic were restrained by these means during the night, restricting their movement. In this case, unfortunately, probably in an attempt to get out of bed, the man became stuck by due to the restraint, accidentally causing strangulation with constriction of the chest. In this case, an analysis was conducted on the role of health personnel and the real need to apply restraints.

In fact, it is shown in the literature that the calculation of the number of nurses/number of patients is essential to reduce the use of physical restrictions on patients [17,18,19]. Furthermore, the use of restraint devices increases with the years of experience of the healthcare personnel and with the reduction in their degree of empathy towards pain [14]. The use of restraints currently seems an even more preponderant problem due to the SARS-CoV-2 pandemic, since the patients hospitalized in these clinics with these disorders are more isolated from relatives and family members and, if they are positive for SARS-CoV-2, they are increasingly isolated by the application of restraints in the absence of custody.

## 6. Conclusions

It is necessary to evaluate the means of restraint as an extraordinary and not ordinary intervention. The choice of the means to be used must be applied to the case and limited in time. Each patient undergoing restraint measures must be carefully monitored by specialized personnel. In the case reported, we highlighted the lack of surveillance of patients undergoing restraint. Therefore, in this case, the lack of patient surveillance and the unjustified application of these means caused an avoidable death. We therefore emphasize the importance of reducing the use of physical restraint, which limits the right to personal freedom of patients and of promoting greater use of environmental and structural restraints [20]. The reported case suggests the need for new fatal-event-prevention protocols. The suggested protocol was compared with the research results, showing the importance of adequate staff training in the use of restraints on patients with mental health disorders and the risk of violent incidents. Furthermore, the safety of restraints and respect for human rights while using them are mandatory [21,22,23].

In this protocol, we propose the following measures:-Provide health personnel with periodic information on the appropriate use of restraints;-Improve operator–patient relationships so that the healthcare staff can personally supervise patients;-Increase the use of devices that allow video surveillance;-Apply safety devices for opening and closing doors;-Apply protective bars to windows to reduce the risk of precipitation;-Apply rails to beds to prevent falls;-Use safety devices in buildings and in their surrounding environments;-Eliminate all objects that could be used by patients as weapons to harm themselves or others (pointed objects, blunt or sharp objects, curtains, carpets).

## Figures and Tables

**Figure 1 ijerph-19-08432-f001:**
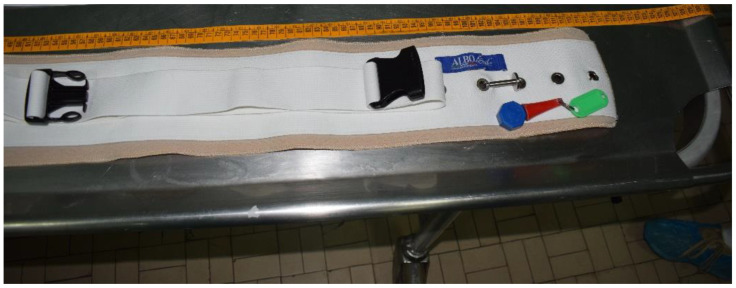
Analysis of restraint belt 115 cm long and 12.5 cm wide.

**Figure 2 ijerph-19-08432-f002:**
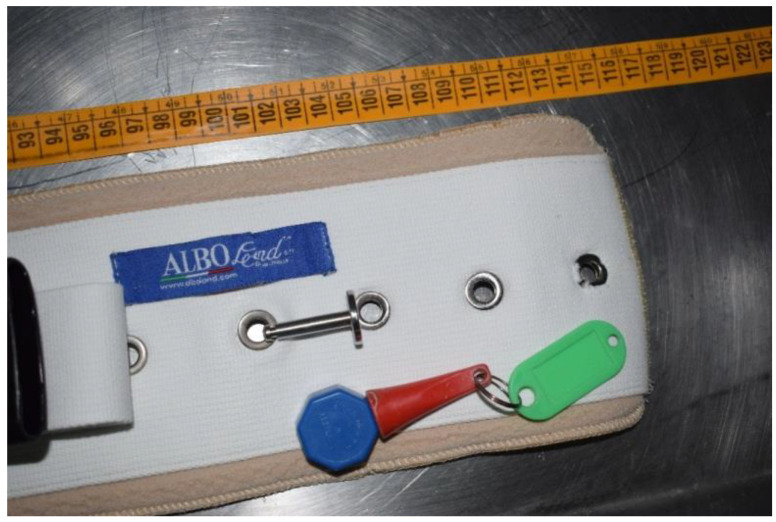
Evaluation of the belt, which was not elastic and had buttonholes.

**Figure 3 ijerph-19-08432-f003:**
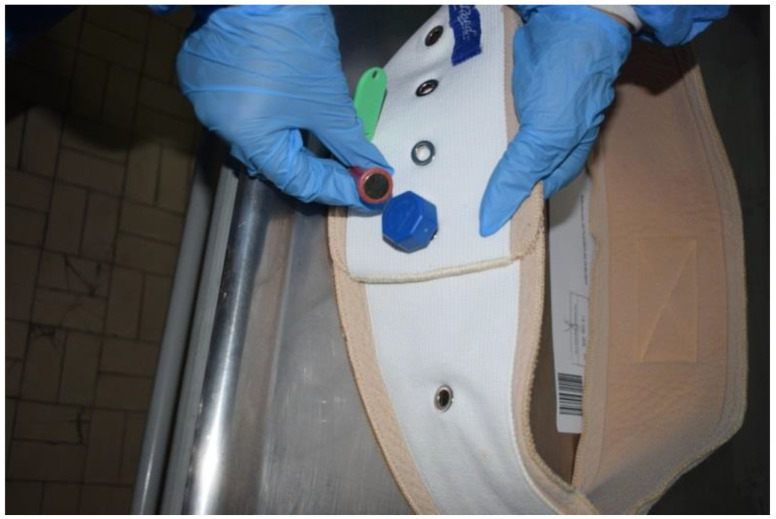
Analysis of the slots with special metal device and magnetic closure.

**Figure 4 ijerph-19-08432-f004:**
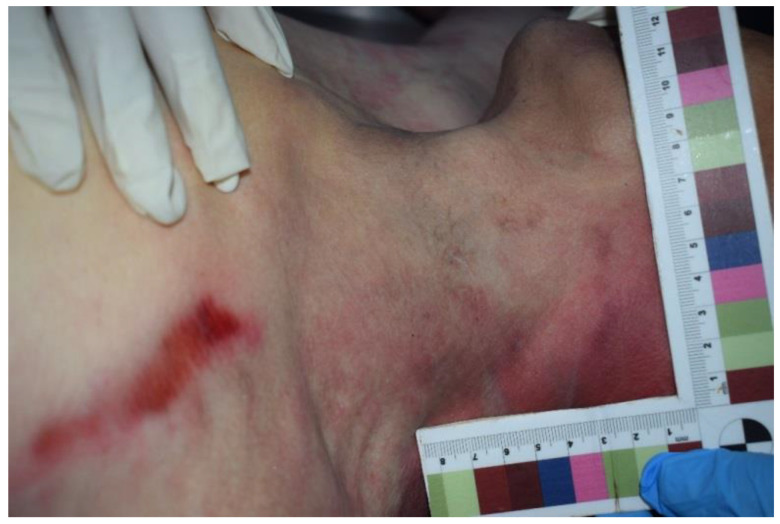
Analysis of the neck with restraint injury.

**Figure 5 ijerph-19-08432-f005:**
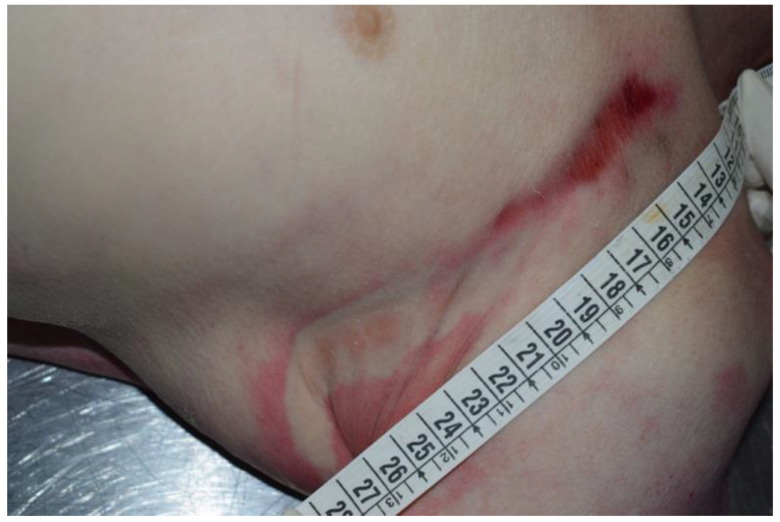
External cutaneous excoriation compatible with sign of restraint.

**Figure 6 ijerph-19-08432-f006:**
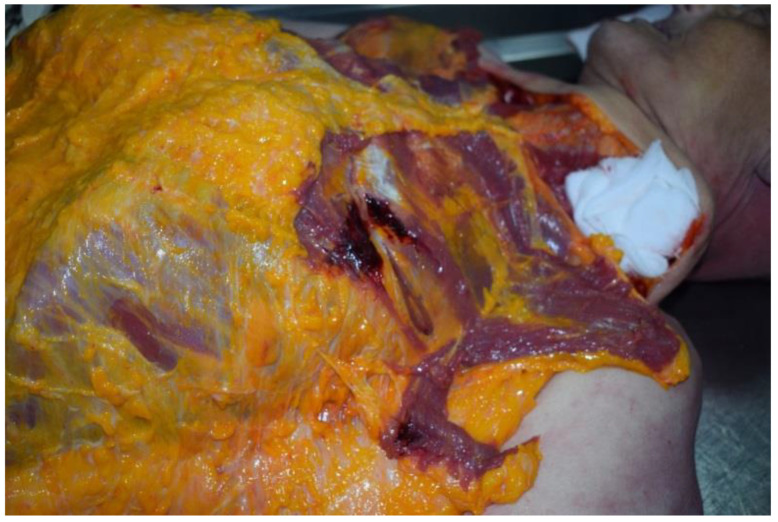
Evaluation of the injury of left intercostal muscles.

## Data Availability

Not applicable to this article as no datasets were generated.

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
