# Peer review of "Hospital Restraints: Safe or Dangerous? A Case of Hospital Death Due to Asphyxia from the Use of Mechanical Restraints"

_ijerph, 2022, doi:10.3390/ijerph19148432_

Round 1

Reviewer 1 Report

The authors report the case of an institutionalized patient died because of physical restraint. The paper is very interesting and the methodology is correct. In particular, it deals with a topic of great relevance for both forensic medicine and public health, since seclusion and restraint for subjects affected by mental illness is a topic highly debated and with substantial implications (do the risks outweight the benefits?). I think that the authors could add few sentences on the issues of seclusion and restraint for subjects affected by mental illness from a medico-legal and public health perspective (thus focusing on the risks of both accidents and suicides). At this regard, some papers could be considered: A systematic review of the effects of prison segregation - 10.1016/j.avb.2020.101389; Suicide of isolated inmates suffering from psychiatric disorders: when a preventive measure becomes punitive - 10.1007/s00414-017-1704-5; Using the stress–vulnerability model to better understand suicide in prison populations - 10.1080/13218719.2021.2013340; The Core Competency Model for Corrections: An Education Program for Managing Self-Directed Violence in Correctional Institutions - 10.1037/ser0000624. Finally, I would add an explanation for lines 81-83 and I would explain what RSAs are (line 142). 

Author Response

Response to Reviewer 1 Comments

Point 1: The authors report the case of an institutionalized patient died because of physical restraint. The paper is very interesting and the methodology is correct. In particular, it deals with a topic of great relevance for both forensic medicine and public health, since seclusion and restraint for subjects affected by mental illness is a topic highly debated and with substantial implications (do the risks outweight the benefits?). I think that the authors could add few sentences on the issues of seclusion and restraint for subjects affected by mental illness from a medico-legal and public health perspective (thus focusing on the risks of both accidents and suicides). At this regard, some papers could be considered: A systematic review of the effects of prison segregation - 10.1016/j.avb.2020.101389; Suicide of isolated inmates suffering from psychiatric disorders: when a preventive measure becomes punitive - 10.1007/s00414-017-1704-5; Using the stress–vulnerability model to better understand suicide in prison populations - 10.1080/13218719.2021.2013340; The Core Competency Model for Corrections: An Education Program for Managing Self-Directed Violence in Correctional Institutions - 10.1037/ser0000624.

Response 1: Thank you very much for your revisions, we really appreciate your comments. The paper has been improved with your suggestions and the proposed references have been analysed and added. The corrections have been underscored in the new manuscript attached. Thank you for your support.

Finally, I would add an explanation for lines 81-83 and I would explain what RSAs are (line 142). 

Response 2: More details have been added and the term RSA has been changed with RCF (Residential Care Facility)

Author Response

Response to Reviewer 2 Comments

Response 1: Thank you very much for your revisions, we really appreciate your corrections. The corrections have been added in text as suggested and they are underscored in the new manuscript attached.

Reviewer 3 Report

This paper reports a case of death due to asphyxia in a Down Syndrome boy. There are some points minor points which will help improve the paper. 

Blockade or restriction of airway could be the cause of death. However, based on the photos provided, the lateral side of the neck especially on the left side (Figure 4) was reddish. This area is where you locate the phrenic nerve. Is it possible that the phrenic nerve was compressed leading to death? How can we rule out such possibility? This point should be discussed more thoroughly.

Age of the victim should be reported in the case report section. 

The authors proposed a protocol to prevent fatalities due to restraint. It was based on which protocols? I believe comparison between the proposed protocol and the existing protocols is necessary. These guidelines should be studied:

https://www.nice.org.uk/guidance/ng10

https://pubmed.ncbi.nlm.nih.gov/11949566

https://www.ncbi.nlm.nih.gov/pmc/articles/PMC6482694/

Author Response

Response to Reviewer 3 Comments

Point 1: This paper reports a case of death due to asphyxia in a Down Syndrome boy. There are some points minor points which will help improve the paper. 

Blockade or restriction of airway could be the cause of death. However, based on the photos provided, the lateral side of the neck especially on the left side (Figure 4) was reddish. This area is where you locate the phrenic nerve. Is it possible that the phrenic nerve was compressed leading to death? How can we rule out such possibility? This point should be discussed more thoroughly.

 Response 1: Thank you very much for your revisions, we really appreciate your comments. The text has been improved with a description of autopsy findings. Vagus nerve injury was found, along with chest compression, even if no injuries on phrenic nerve was found.

Point 2: Age of the victim should be reported in the case report section. 

  Response 2: The age (60 years old) was added.

Point 3: The authors proposed a protocol to prevent fatalities due to restraint. It was based on which protocols? I believe comparison between the proposed protocol and the existing protocols is necessary. These guidelines should be studied:

https://www.nice.org.uk/guidance/ng10

https://pubmed.ncbi.nlm.nih.gov/11949566

https://www.ncbi.nlm.nih.gov/pmc/articles/PMC6482694/

Response 3: A comparison with suggested protocols was added and the suggested guidelines have been added in references.

Reviewer 4 Report

Dear Authors, your case reports is very interesting and adds to the forensic literature. In general I think you have covered all the points that deemed to be  made. I would only like to ask you to add something (once sentence or short paragraph, with 1-2 references) about the use of restraint methods in the psychiatric setting in Italy: is it customary? Was it always like that? Are there specific laws regulating it?

Author Response

Response to Reviewer 4 Comments

Point 1: Dear Authors, your case reports is very interesting and adds to the forensic literature. In general I think you have covered all the points that deemed to be  made. I would only like to ask you to add something (once sentence or short paragraph, with 1-2 references) about the use of restraint methods in the psychiatric setting in Italy: is it customary? Was it always like that? Are there specific laws regulating it?

Response 1: Thank you very much for your revisions, we really appreciate your comments. The paper has been improved with your suggestions and a part about italian law about restraints has been added. The corrections have been underscored in the new manuscript attached. Thank you for your support.